# Cytoplasmic genetic variation and extensive cytonuclear interactions influence natural variation in the metabolome

Bindu Joseph[1], Jason A Corwin[1], Baohua Li[1], Suzi Atwell[1], Daniel J Kliebenstein[1,2]*

[1]Department of Plant Sciences, University of California, Davis, Davis, United States; [2]DynaMo Center of Excellence, University of Copenhagen, Frederiksberg, Denmark

**Abstract** Understanding genome to phenotype linkages has been greatly enabled by genomic sequencing. However, most genome analysis is typically confined to the nuclear genome. We conducted a metabolomic QTL analysis on a reciprocal RIL population structured to examine how variation in the organelle genomes affects phenotypic variation. This showed that the cytoplasmic variation had effects similar to, if not larger than, the largest individual nuclear locus. Inclusion of cytoplasmic variation into the genetic model greatly increased the explained phenotypic variation. Cytoplasmic genetic variation was a central hub in the epistatic network controlling the plant metabolome. This epistatic influence manifested such that the cytoplasmic background could alter or hide pairwise epistasis between nuclear loci. Thus, cytoplasmic genetic variation plays a central role in controlling natural variation in metabolomic networks. This suggests that cytoplasmic genomes must be included in any future analysis of natural variation.

## Introduction

A central goal of modern biology is to understand how the integration of gene functions across a genome lead to the individual's specific phenotype. A key facet to this effort is to develop models that would allow directly inferring a species phenotypic variation from its genetic variation. This goal of mathematically linking genetic to phenotypic variation is central to all studies of quantitative genetics ranging from human genetics to plant breeding to ecology and has led to massive genome resequencing projects focused on developing the genomic databases to allow these studies (*Liti et al., 2009*; *Altshuler et al., 2010*; *Cao et al., 2011*). However, in most quantitative genomics studies, genome analysis is largely confined to the nuclear genome with much less attention paid to the organellar genome. This is in contrast to the central role that the organellar genome plays in controlling organismal metabolism and function and the developing body of literature suggesting that organellar genomic variation can modulate the effects of nuclear genomic variation.

Genomic variation in human organelles has been linked to several severe diseases (*Wallace et al., 1988*; *Taylor and Turnbull, 2005*; *Schon et al., 2012*). However, these organellar variants are typically rare with large phenotypic consequences such that they can be followed using simple maternal inheritance studies without consideration for quantitative variation in the nuclear genome (*Schon et al., 2012*). More recently, quantitative studies on human diseases suggest that genetic variation in organellar genomes modify the quantitative effect of nuclear loci and disease phenotypes (*Battersby and Shoubridge, 2001*; *McRae et al., 2008*; *Schon and Przedborski, 2011*). Additionally, the use of structured populations in mice, yeast, and birds have shown that cytoplasmic genome variation can influence high order phenotypes including fitness, cognition, and biomass (*Roubertoux et al., 2003*; *Zeyl et al., 2005*; *Park et al.,*

*For correspondence:
kliebenstein@ucdavis.edu

Joseph *et al*. eLife 2013;2:e00776. DOI: 10.7554/eLife.00776

**eLife digest** The vast majority of genes in plant and animal cells are located on chromosomes within the nucleus. However, cells also contain a small number of genes outside the nucleus in cellular organelles such as the mitochondria, which generate energy, and the chloroplasts, which carry out photosynthesis. All these non-nuclear genes comprise the organellar genome.

When trying to explain how variation in genes leads to differences in the characteristics of animals and plants, geneticists have historically paid most attention to the genes inside the nucleus. However, more recent work has shown that variation in the organellar genome can also contribute to differences between individuals, although the relative contribution of organellar genes versus nuclear genes remains unclear.

Now, Joseph et al. have performed the first large-scale analysis of how variation in the organellar genome affects the characteristics (or phenotype) of the plant model organism, Arabidopsis. The study examined the degree to which variation in each of roughly 13,000 nuclear genes and 200 organellar genes affected the levels of thousands of metabolites inside cells.

This metabolomics analysis revealed that variation in the organellar genome contributed to variation in the levels of more than 80% of the metabolites studied. Organellar genes also helped to regulate the effect of nuclear genes. This combination of direct and indirect influences helps to explain how a small number of organellar genes can have a disproportionately large effect on phenotype.

The work of Joseph et al. suggests that the role of the organellar genome has been significantly underestimated to date, and that geneticists should consider variation in both the nuclear and organellar genome when attempting to determine how genes affect phenotype.

2006; *Dimitrov et al., 2009*). Within ecological studies, mitochondrial variation has also been shown to alter fitness and create hybrid isolation in a variety of invertebrate species (*Willett and Burton, 2004*; *Wade and Goodnight, 2006*; *Dowling et al., 2007, 2010*; *Wolf, 2009*; *Willett, 2012*). However, these studies didn't directly interrogate the interaction of the nuclear and cytoplasmic genomes for quantitative traits or test the breadth of phenotypes affected.

In plants, genetic variation in mitochondria is also linked with large qualitative phenotypes, such as cytoplasmic male sterility (*Hanson, 1991*; *Schnable and Wise, 1998*). Plant breeding has a long history of using diallele crosses to test for the presence of maternal effects (cytoplasmic genetic variation) on a phenotype and this was recently extended to a small reciprocal F2 family structure to show that the cytoplasmic effects could have significant impacts on plant height in maize (*Tang et al., 2013*). In rice, agronomic traits have also been shown to be influenced by interactions between cytoplasmic and nuclear genomes but the specific loci were not identified (*Tao et al., 2004*). However, these reciprocal F2 populations typically have generated the impression that cytoplasmic effects on phenotypic variation are quite small in plants possibly because of an inability to account for interactions between the nuclear and cytoplasmic genomes (*Singh, 1965*; *Crane and Nyquist, 1967*; *Eenink and Garretsen, 1980*; *Miura et al., 1997*; *Primomo et al., 2002*). In contrast to the previous estimates of small effects, genomic sequencing within Arabidopsis has shown the presence of considerable genetic polymorphism in both the plastidic and mitochondrial genomes suggesting the potential for broad phenotypic consequences (*Moison et al., 2010*).

The above studies have shown that cytoplasmic genome variation can influence phenotypic variation. Additionally, genes underlying qualitative interactions between the nucleus and cytoplasm leading to cytoplasmic male sterility and interspecific isolation have been identified. There are, however, numerous open questions remaining about how cytoplasmic variation influences quantitative phenotypic variation. What is the level of phenotypic variation influenced by genetic variation in the cytoplasm in comparison to individual nuclear loci? How much epistatic interaction is there between genetic variation in nuclear loci and the cytoplasmic genomes? What is the breadth of phenotypes that might be influenced by cytoplasmic variation?

To begin answering these questions, we utilized metabolomics to investigate how genetic variation in the cytoplasmic and nuclear genomes interacts to control metabolome variation in the reciprocal Arabidopsis Kas × Tsu recombinant inbred line (RIL) population (*McKay et al., 2008*; *Juenger et al.,*

*2010*). We focused on metabolomics because it is cost effective for large sample numbers and because plant organelles, mitochondria and plastids, are central to the function of plant primary metabolism, as well as many specialized metabolites in both energy generation and biosynthesis (*Fiehn et al., 2000*; *Roessner et al., 2001*; *Fiehn, 2002*). Within Arabidopsis, there are nearly 13,000 genes in the nuclear genome predicted to be involved in metabolic processes of which ~3000 may be targeted to the mitochondria and the plastid (*Arabidopsis genome initiative, 2000*). This included complete biosynthetic pathways for numerous amino acids and key energy production processes like respiration and photosynthesis. Aiding these processes are 88 total predicted genes in the plastidic genome and 121 in the mitochondrial genome; most of which function to facilitate the metabolic processes that occur in organelles using both nuclear and cytoplasmic-encoded proteins. Thus, variation in the organellar genomes could directly influence the function of any of nuclear gene functioning within the organelle (*Etterson et al., 2007*; *Tang et al., 2007*; *Wolf, 2009*). This suggests that metabolism is an ideal phenotype to use for testing how cytoplasmic variation can influence phenotypic variation.

We measured the metabolome within the Kas × Tsu RIL population to test how cytoplasmic variation can affect quantitative variation within the metabolome. This population was generated from a reciprocal cross and approximately half of the resulting lines carry Kas organelles while the other half carry Tsu organelles allowing for explicit analysis of the influence of cytoplasmic genetic variation (*McKay et al., 2008*; *Juenger et al., 2010*). Analysis of the Kas × Tsu RIL population metabolome showed that genetic variation in the organelles influenced the accumulation of over 80% of the detectable metabolites. In contrast to previous observations suggesting that the cytoplasm has only small effects, phenotypic changes associated with cytoplasmic variation were as large and often larger than that found for individual nuclear loci (*Singh, 1965*; *Crane and Nyquist, 1967*; *Eenink and Garretsen, 1980*; *Miura et al., 1997*; *Primomo et al., 2002*). In addition, the cytoplasm was found to be a central hub in the epistatic network controlling natural variation in plant metabolism. This centrality led to the cytoplasmic background displaying the unexpected ability to hide or alter epistatic interactions between nuclear loci. Thus, genetic variation in the cytoplasmic organelles has widespread and large quantitative effects on natural phenotypic variation and can influence the link between nuclear loci.

## Results

### Comparative metabolome genetics across populations

To partition phenotypic variance between the effects of the nuclear and cytoplasmic genomes upon the *A. thaliana* metabolome, we identified and measured metabolite levels, using non-targeted GC-TOF-MS, in leaf tissues of 316 lines of the Kas × Tsu *A. thaliana* RIL population harvested with fourfold replication across two separate experiments. Within the 316 lines for this population, 136 have the Kas cytoplasm and 180 have the Tsu cytoplasm. A total of 2435 metabolites were identified in over 25% of the RILs with 215 of these being known compounds and 2220 being unidentified compounds. Of these 2435 metabolites, 559 were identified in both experiments (161 known and 398 unknown) while the rest were specific to one of the two experiments.

Classical breeding studies utilize reciprocal crosses with linear modeling to partition heritability between the nuclear and cytoplasmic genomes (*Singh, 1965*; *Crane and Nyquist, 1967*; *Eenink and Garretsen, 1980*; *Miura et al., 1997*; *Primomo et al., 2002*). In line with the standard heritability calculations for classical breeding experiments with reciprocal crosses, we used a linear modeling approach to approximate the level of heritability in this population that can be statistically ascribed to the nuclear and cytoplasmic genome. The line heritability model simply partitions the RILs into two subpopulations based on the maternal parent and tests the level of reproducibility that is controlled by differences between the lines and between the subpopulations. Using the line heritability model, we focused on the 559 metabolites found in both experiments to estimate the broad sense heritability of the metabolome. This analysis showed that 361 metabolites had a significant (p<0.01) line effect, of which 334 metabolites showed significant heritability based on the maternal parent subpopulation. This maternal effect was quite small with an average of 1.6 ± 0.1% (range 15–0%) (*Figure 1* and *Figure 1—source data 1*). In comparison, 77 metabolites had significant nuclear broad sense heritability with an average of 21.3 ± 0.3% (range 58–2%) (*Figure 1* and *Figure 1—source data 1*). This is a similar level of variation to that found in a previous analysis of metabolomic variation in Arabidopsis (*Rowe et al., 2008*; *Chan et al., 2010*). The combined variance of the genetic components, both nuclear and cytoplasmic, explained approximately 19 times as much of the variance as the combination of experiment

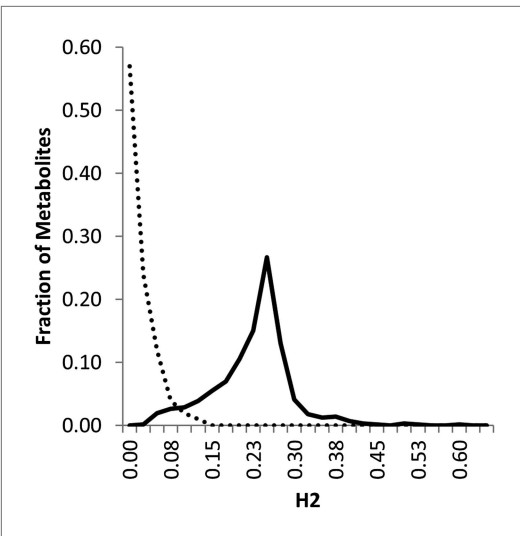

**Figure 1**. Line estimation of heritability in nuclear and organellar genomes. We compared the estimated metabolite heritability's due to nuclear (solid line) and organelle (dashed line) variation across the lines of the Kas × Tsu (black) RIL populations. Shown are frequency plots of heritability and for each class, the bin size is 5% for the frequency plots.
The following source data are available for figure 1:

**Source data 1**. Heritability.
**Source data 2**. Means.

and block (*Figure 1—source data 1*) (*Rowe et al., 2008*; *Chan et al., 2010*). This raises an apparent disparity between the nuclear genome having high heritability that is significant for a low fraction of metabolites (*Figure 1—source data 1*). In contrast, the cytoplasmic effects have relatively low average heritability but significantly impact most metabolites (*Figure 1—source data 1*). The most likely explanation for this disparity is the large difference in the degrees of freedom for the two factors, nuclear vs cytoplasmic used for the significance test. When conducting the ANOVA test, there are only two maternal subpopulations, Kas or Tsu, whereas there are 316 different nuclear genomes/RILs, and as such, the F value for the same level of variance will be very different in the two tests (*Rowe and Kliebenstein, 2008*). Thus, variation in most metabolites identifies a significant effect of cytoplasmic genetic variation, even though this effect appears to be smaller than that for the nuclear genome in the standard line heritability estimation approach.

## Single locus analysis of nuclear variation and the plant metabolome

We next moved beyond the classical line heritability approach and directly test if cytoplasmic variation may have similar effects to individual nuclear loci. We mapped QTLs in the Kas × Tsu RILs controlling the mean accumulation of all 2435 measured metabolites (215 known and 2220 unidentified compounds) using 1069 markers with a median spacing of 0.35 cM (*Figure 1—source data 2*) (*Jansen, 1994*; *Zeng et al., 1999*; *Broman et al., 2003*). This identified 2974 QTLs effecting the accumulation of 1822 metabolites with an average of 1.22 ± 0.02 QTLs per metabolite (*Figure 2—source data 1*). These QTLs predominantly partitioned into 14 metabolomic QTL hotspots that were unequally distributed across the genome (*Figure 2*). Chromosome IV had five detectable hotspots while chromosomes I and II had only a single significant hotspot (*Figure 2*). The hotspots had an average LOD interval size of approximately 5 cM that on average contains a similar number of genes as found within the combined mitochondrial and plastidic genomes.

To compare nuclear and cytoplasmic genetic variation effects upon metabolism, we utilized these metabolomic QTL hotspots to develop an additive model. This model uses the genetic marker at the center of each metabolomic QTL hotspot as separate terms and also incorporates the cytoplasmic variation as an additional term equivalent to each nuclear locus. We used this additive model to directly test all QTL hotspot-metabolite linkages and obtain the mean effect of variation at each hotspot on the metabolome (*Figure 3—source data 1*, *Figure 3—Figure supplements 1–12*). For all hotspots, the primary metabolites altered by variation at each hotspot were distributed across the primary metabolism pathways with no obvious strong qualitative network specific loci as previously found (*Figure 3*) (*Rowe et al., 2008*).

## The comparative effects of cytoplasmic genome variation to nuclear variation

We next proceeded to use the additive model to directly compare the role of cytoplasmic genome variation to that of individual nuclear loci in controlling metabolome variation (*Figure 3—source data 1*). In comparison to the broader question of cytoplasmic heritability tested in the line heritability model (*Figure 1—source data 1*), this model allows us to ask the more specific question of how the cytoplasmic

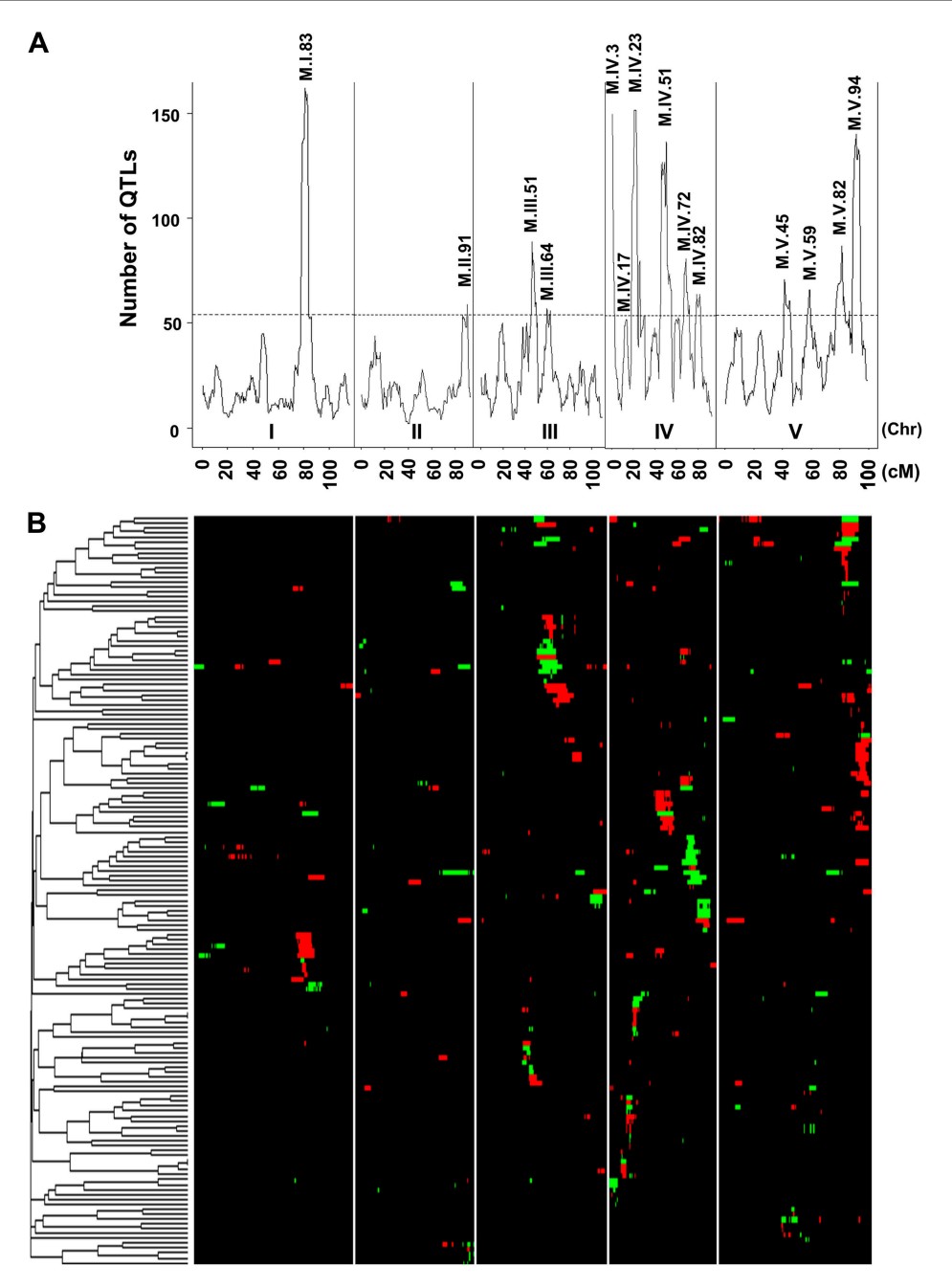

**Figure 2**. Genetic architecture of metabolite QTLs across the Kas × Tsu genome. (**A**) The number of metabolites for which a QTL was detected within a 5 cM sliding window is plotted against the genetic location of the metabolite QTLs in cM. The permuted threshold (p=0.05) for detection of a significant metabolite hotspot is 54 QTLs. The graph is scaled to match part (**B**). Hotspots are labeled above the respective locus with the chromosome and cM. (**B**) Heat map showing the location and effect of significant QTLs detected for average metabolite accumulation across the five chromosomes. Red indicates a positive effect of the Kas allele, while green indicates a positive effect of the Tsu allele. Vertical white lines separate the chromosomes (I to V from left to right). Clustering on the left is based on the absolute Pearson correlation of QTL effects across all significant loci for each metabolite. Only metabolites with two or more QTLs were plotted.

The following source data are available for figure 2:

**Source data 1**. QTL Lists.

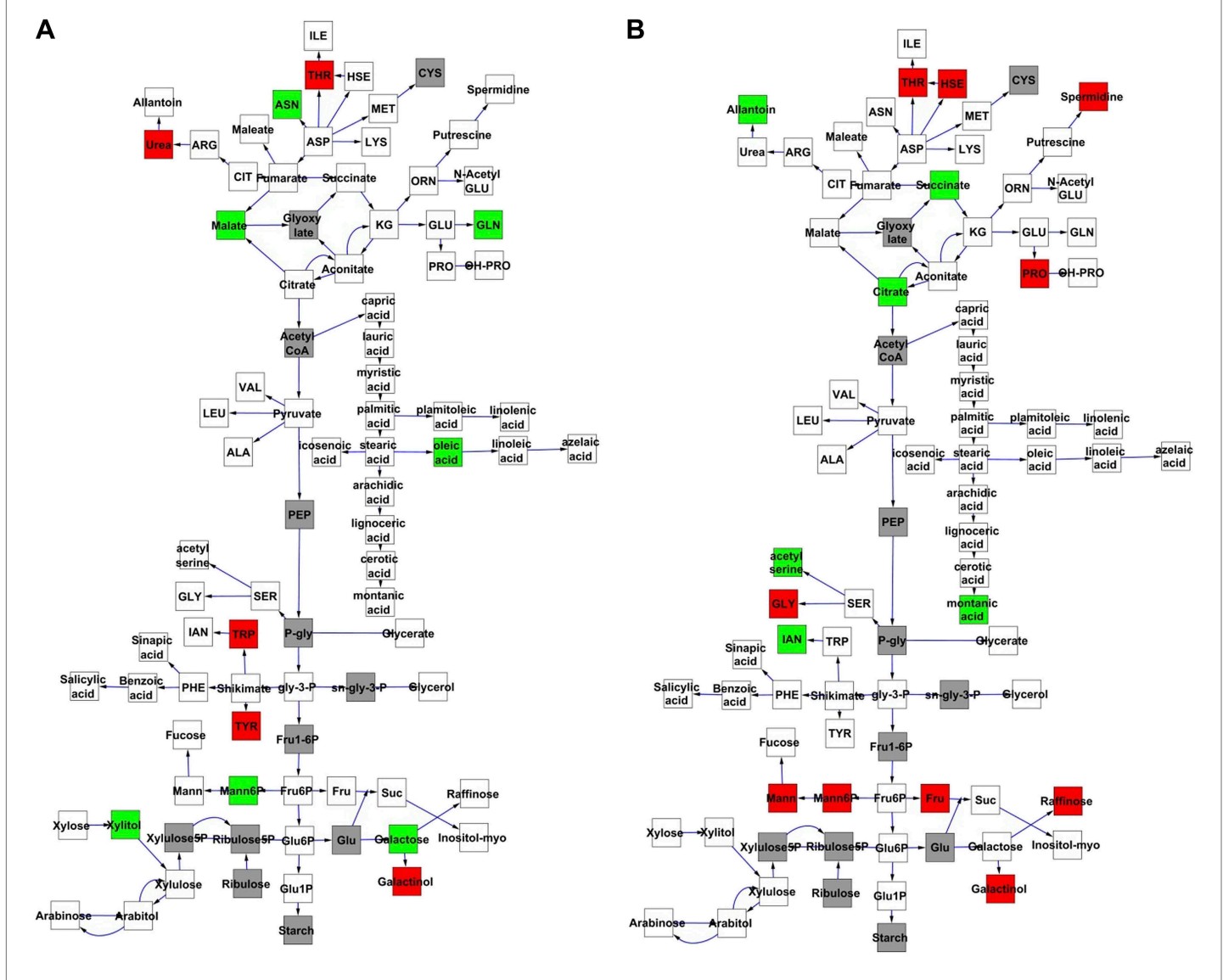

**Figure 3**. Metabolomic consequence of variation at nuclear loci. A map of central metabolism was created in cytoscape and this was used to plot the estimated allele effect of genetic variation at nuclear loci. A red box shows increased metabolite accumulation when the line contains the Kas allele at the nuclear locus while green shows increased metabolite accumulation when the line contains the Tsu allele at the nuclear locus. White boxes are metabolites that were detected but not significantly influenced by the cytoplasmic genome and gray boxes are metabolites that were not detected. The two loci shown are those that had the most metabolites affected within the metabolic map. All other nuclear loci are plotted in *Figure 3—figure supplements 1–12*. (**A**) Estimated allele effects of the M.V.59 hotspot. (**B**) Estimated allele effects of the M.I.83 hotspot.

The following source data and figure supplements are available for figure 3:

**Source data 1**. Single marker ANOVA.

**Figure supplement 1**. Metabolomic consequence of variation at nuclear locus M.II.91.

**Figure supplement 2**. Metabolomic consequence of variation at nuclear locus M.III.51.

**Figure supplement 3**. Metabolomic consequence of variation at nuclear locus M.III.64.

*Figure 3. Continued on next page*

*Figure 3. Continued*

**Figure supplement 4**. Metabolomic consequence of variation at nuclear locus M.IV.3.

**Figure supplement 5**. Metabolomic consequence of variation at nuclear locus M.IV.17.

**Figure supplement 6**. Metabolomic consequence of variation at nuclear locus M.IV.23.

**Figure supplement 7**. Metabolomic consequence of variation at nuclear locus M.IV.51.

**Figure supplement 8**. Metabolomic consequence of variation at nuclear locus M.IV.72.

**Figure supplement 9**. Metabolomic consequence of variation at nuclear locus M.IV.82.

**Figure supplement 10**. Metabolomic consequence of variation at nuclear locus M.V.45.

**Figure supplement 11**. Metabolomic consequence of variation at nuclear locus M.V.82.

**Figure supplement 12**. Metabolomic consequence of variation at nuclear locus M.V.94.

variation compares to individual nuclear loci in altering metabolite accumulation. The additive model showed that variation in the cytoplasmic genome significantly altered variation in 1755 out of the 2435 detected metabolites. This is in comparison the nuclear loci that only affected 298 metabolites on average (range 438–158). In addition to affecting more metabolites than nuclear loci, the average effect size of the cytoplasmic genome on metabolite variation was higher than that for the nuclear loci (*Figure 4*).

Using the additive model, we generated per locus estimates of heritability that showed the cytoplasmic genome explained twice the variance as any individual nuclear locus on average (*Figure 4—figure supplements 1–8*). Interestingly, the number of genes within a 5 cM region surrounding the nuclear hotspots is comparable to the number of genes present within the combined organellar genomes, which suggests that this is not simply a matter of genetic potential. Additionally, the Kas allele of both the nuclear and cytoplasmic loci were more frequently associated with increasing metabolite concentration leading to a change in the entire metabolic network within this population based on the cytoplasmic genome (*Figures 4 and 5*). Thus, the line heritability approach to partitioning genetic variation dramatically underestimated the effect of cytoplasmic genetic variation and cytoplasmic variation has larger effects than any individual nuclear locus even though there are a similar potential number of causative genes.

## Cytoplasmic polymorphisms

To identify potential causal polymorphisms between the Kas and Tsu organelles, we obtained short-read sequencing data and identified SNPs in the organellar genomes. This analysis showed that there were polymorphisms spread out across a large number of genes including regulatory genes, such as *rpoC2* in the plastid, several unknown genes, and key energy genes, such as Rubisco large subunit. The most striking changes were the large number of polymorphisms within the NADH deyhdrogenases in both the mitochondria and the plastid (*Figure 5—source data 1*). Specifically in the mitochondria, 32 of the 96 polymorphisms were within the NADH dehydrogenase 7 and a further 13 polymorphisms in four other NADH dehydrogenase complex genes. This is vastly more polymorphisms than would be expected by random chance suggesting a change in NADH and related metabolism between the Kas and Tsu organelles (hyper-geometric test, p<0.001). This agrees with the widespread metabolic consequences of the cytoplasmic genetic variation on processes that require NADH like glutamine synthesis, lipid metabolism, and any other process that utilizes NADH requiring cytochromes P450. However, within this population the mitochondrial and plastidic genomes are perfectly co-inherited making it impossible to resolve the individual effects of the mitochondrial and plastidic genomes (*Joseph et al., 2013*). Additionally, it isn't possible to ascribe metabolites to specific subcellular compartments using a whole tissue extract as most metabolites, even those synthesized in specific

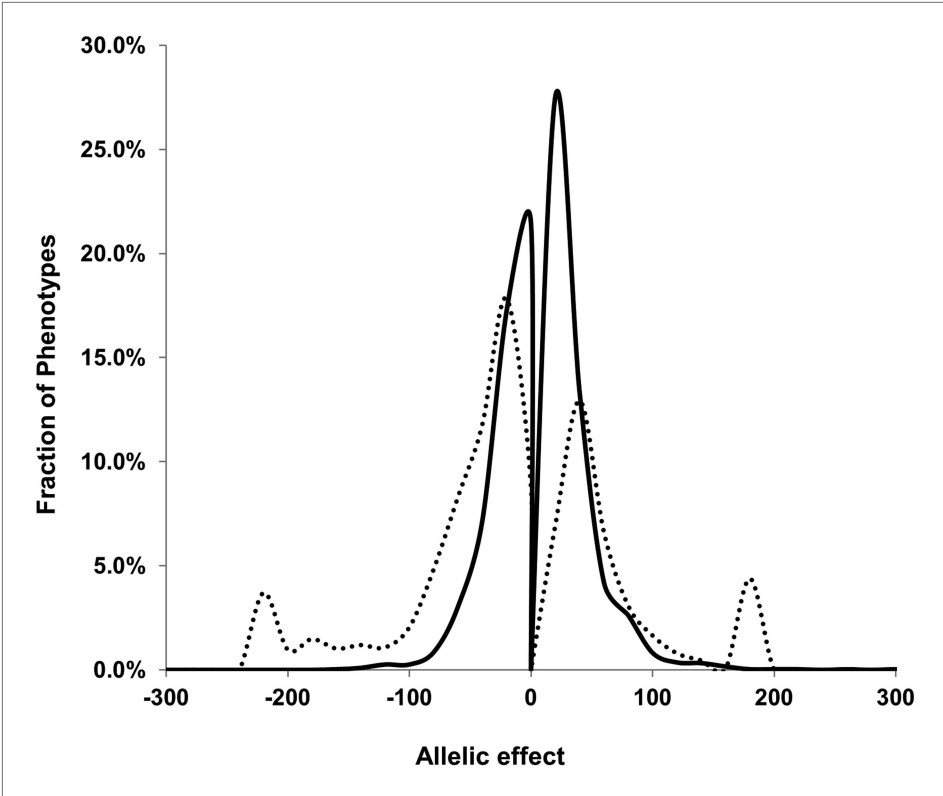

**Figure 4**. Comparison of estimated QTL allele effects between nuclear and cytoplasmic genetic variation. The distribution of percent allele effects are shown for all metabolite/loci combinations with positive being the Tsu allele increases the metabolite concentration in comparison to the Kas allele. Solid black line shows the allele effects for all nuclear genomic loci while the dashed line shows cytoplasmic genetic variation allele effects.

The following figure supplements are available for figure 4:

**Figure supplement 1**. Main effect estimations.

**Figure supplement 2**. Distribution of heritability of pairwise interaction of nuclear loci M.II.91 and M.I.83.

**Figure supplement 3**. Distribution of heritability of pairwise interaction of nuclear loci M.III.64 and M.III.51.

**Figure supplement 4**. Distribution of heritability of pairwise interaction of nuclear loci M.IV.23 and M.IV.17.

**Figure supplement 5**. Distribution of heritability of pairwise interaction of nuclear loci M.IV.51 and M.IV.23.

**Figure supplement 6**. Distribution of heritability of pairwise interaction of nuclear loci M.IV.82 and M.IV.72.

**Figure supplement 7**. Distribution of heritability of pairwise interaction of nuclear loci M.V.59 and M.V.45.

**Figure supplement 8**. Distribution of heritability of pairwise interaction of nuclear loci M.V.94 and M.V.82.

compartments, often accumulate in most compartments within a plant cell (**Krueger et al., 2011**). Thus, genetic variation in the cytoplasm can have significant effects on modulating global plant metabolism requiring further work to identify the specific mechanistic causes.

## Cytonuclear epistasis shapes the plant metabolome

Previous work on natural variation in the plant metabolome had shown extensive two-way and three-way epistasis between nuclear loci but could not test for the presence of epistatic interactions between

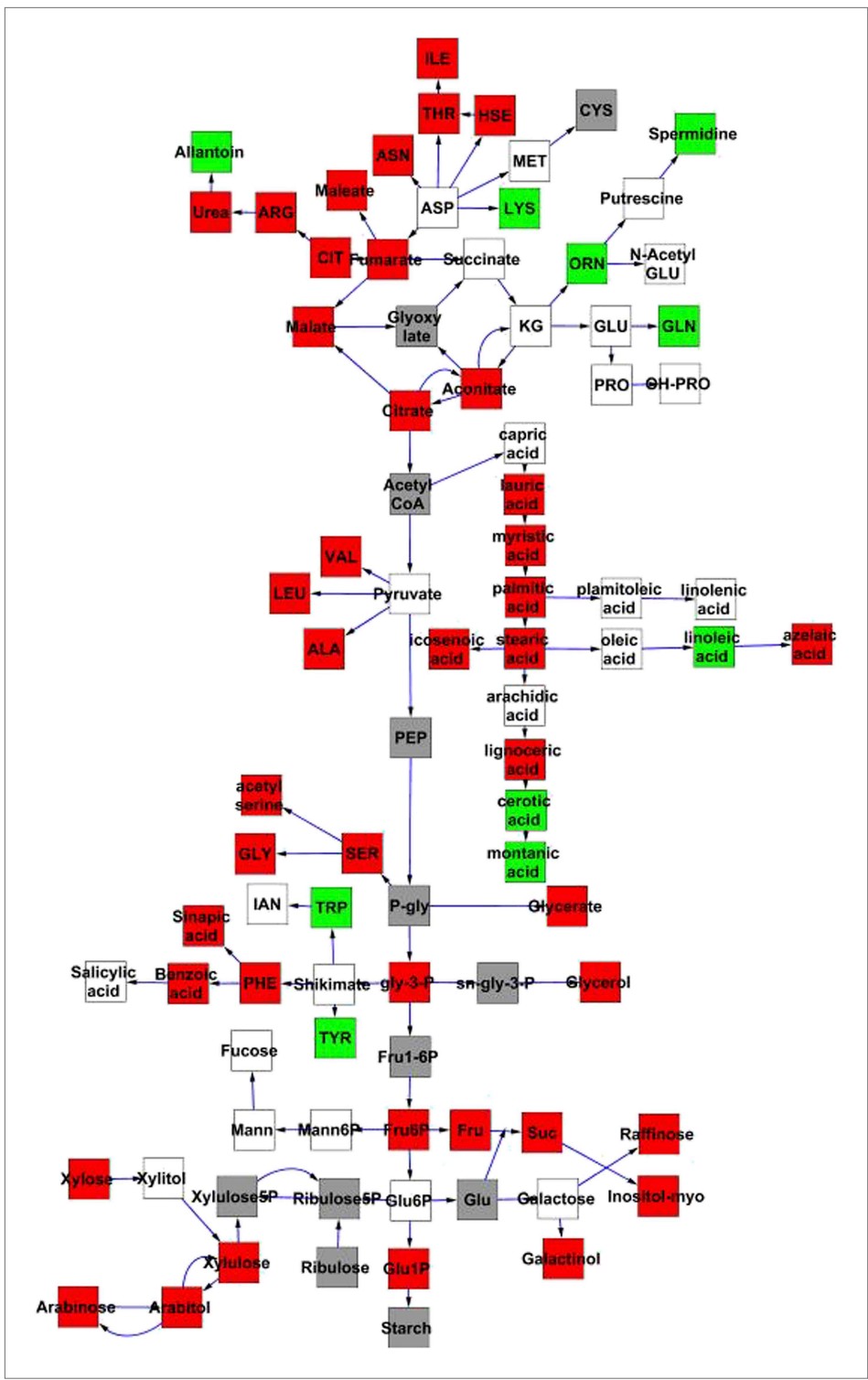

**Figure 5**. Metabolomic consequences of cytoplasmic genomic variation. A map of central metabolism was created in cytoscape and this was used to plot the estimated allele effect of genetic variation in the cytoplasmic genomes using the reciprocal sub-populations. Colors are as given in *Figure 3*.

The following source data are available for figure 5:

**Source data 1**. Genetic polymorphisms between the Kas and Tsu Organelles.

nuclear and cytoplasmic loci (*Rowe et al., 2008*). To test for cytonuclear epistatic interactions, we used a pairwise epistasis model that directly tested all pairwise combinations of the 14 nuclear QTL and cytoplasm against all metabolites while including all terms from the additive model (*Figure 6—source data 1, 2*). This pairwise epistasis model only uses 120 of the 314 available degrees of freedom (38%) leaving the majority of the degrees of freedom for the residual variance, which suggests that we are not over fitting the model. We did not extend this analysis to a full genome survey of all possible loci because these surveys do not account for existing main effect loci. After multiple testing adjustments, we only plotted pairwise interactions that significantly affected at least 10% of the metabolites to present a conservative image of the interaction network (*Figure 6*).

Importantly, the cytoplasmic locus was one of the two nodes with the highest number of significant pairwise epistatic interactions connecting to 6 of the 14 nuclear loci. This level of interaction is higher than the average of the network degree distribution (3.5 ± 0.46 SE) (*Figure 6*). The M.I.83 nuclear locus had the highest level of interactions involving 7 of the other 13 nuclear loci but did not interact directly with the cytoplasmic variation (*Figure 6*). Thus, there is strong pairwise epistasis between the cytoplasmic and nuclear genetic loci. Further, the percent of genetic variation controlled by interactions of two nuclear loci was of similar scale to that involving a cytonuclear interaction (*Figure 4—figure supplements 1–8*).

## Cytoplasmic genetic variation has larger phenotypic effects on metabolite epistasis than the nuclear partner

The large number of significant epistatic interactions between nuclear and cytoplasmic genetic variation forms a cohesive network but the network approach does not let us visualize or directly quantify the effects of each locus in an epistatic pair. Thus, we wanted to develop a better approach to visualize

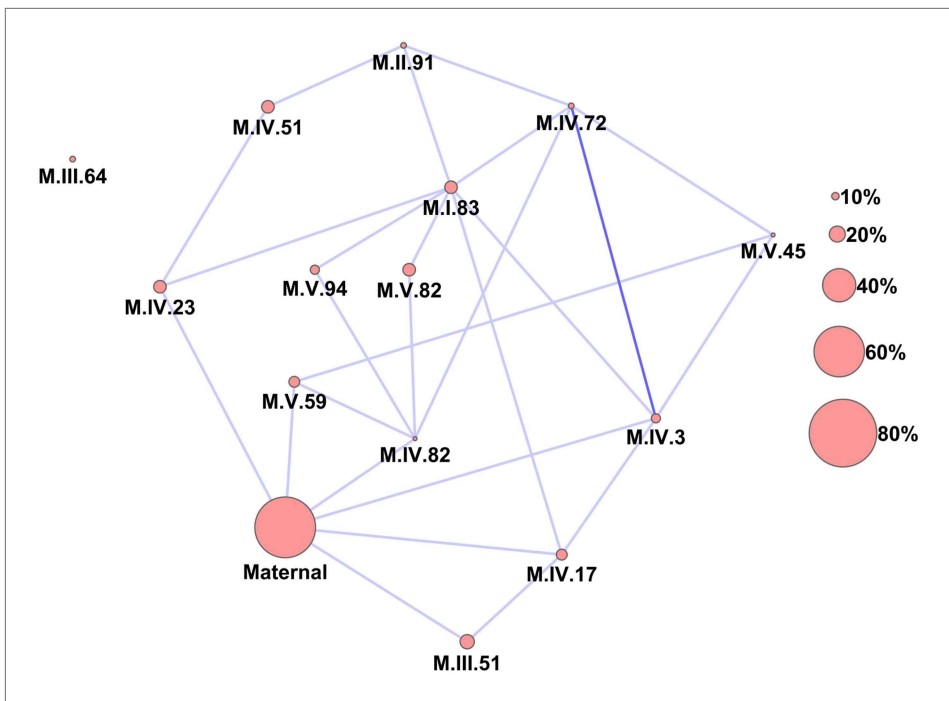

**Figure 6**. Epistatic networks of metabolism. All epistatic interactions between the cytoplasmic genomic variation and nuclear loci (as labeled in *Figure 2*) are plotted as lines connecting the main effect loci as nodes. The size of the main effect nodes represents the fraction of total metabolites affected by the given locus. The color of the lines show the fraction of metabolites linked with this interaction, light blue 10–15% of metabolites, dark blue 15–20% of metabolites.

The following source data are available for figure 6:

**Source data 1**. Pairwise ANOVA p values.

**Source data 2**. Pairwise ANOVA sums of squares.

and compare epistatic effects (*Carlborg and Haley, 2004*; *Carlborg et al., 2006*; *Alvarez-Castro and Carlborg, 2007*). We utilized the center of mass concept from physics to generate a description of epistatic effect (*Gartenhaus and Schwartz, 1957*). In center of mass calculations across two dimensions, each individual objects mass is described in its x,y coordinates. For describing epistasis, we converted the x,y-cartography such that the alleles for one QTL with Tsu being -1 and Kas being 1 are along the x-axis. Similarly, along the y-axis are the alleles at the second QTL thus creating a calculable genotypic cartography (*Figure 7A*). This effectively replaces the physical distance metric in the center of mass calculation with a genetic distance metric such that they represent a scaled allelic effect. We

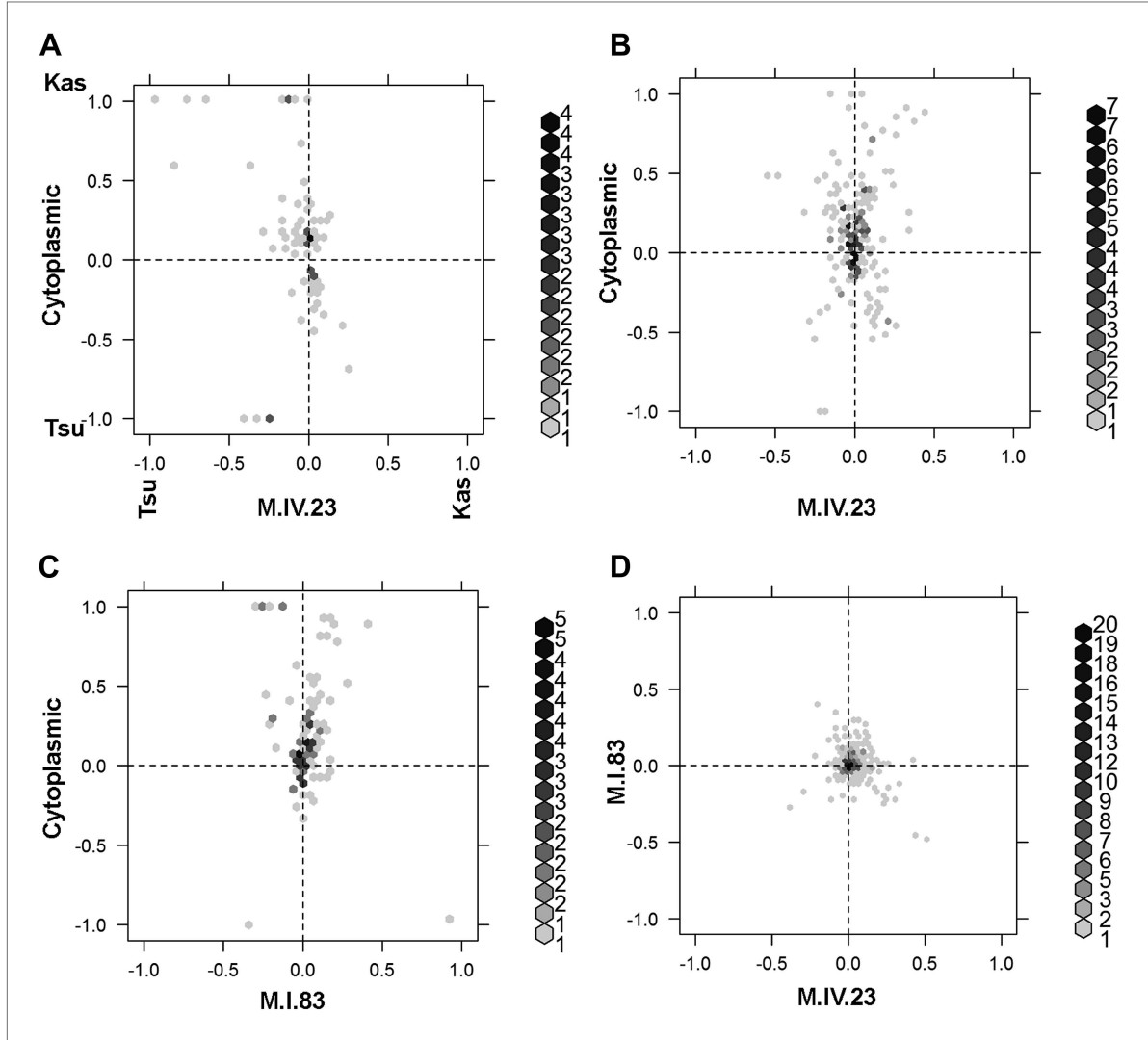

**Figure 7**. Different epistatic patterns across genotypic combinations. The center of mass calculations were used to estimate the phenotypic center for each metabolite that was significantly affected by the given combination of loci. The hexbin plots show the distribution of phenotypic centers for all significant metabolites. The number of metabolites per hexbin are shown in the legend to the right of each graph. All other significant epistatic pair combinations are plotted in *Figure 7—figure supplement 2*. (**A**) Metabolites that are additive for both the cytoplasmic loci and M.IV.23. (**B**) Metabolites that are epistatically affected by an interaction between M.IV.23 and the cytoplasmic genotypes. (**C**) Metabolites that are epistatically affected by an interaction between M.I.83 and the cytoplasmic genotypes. (**D**) Metabolites that are epistatically affected by an interaction between M.I.83 and M.IV.23 genotypes.

The following figure supplements are available for figure 7:

**Figure supplement 1**. Descriptive model of the epistatic center plot.

**Figure supplement 2**. Hexbin plots show the distribution of phenotypic centers for cytoplasmic and nuclear pairwise epistatic interactions.

then replace the individual objects mass the in center of mass calculations with the unscaled average phenotypic value for each of the homozygous genotypic combinations of the two QTLs allowing a genetic centroid of the phenotype to be estimated for each metabolite and plotted (*Figure 7A*). This allows a single plot to compare the magnitude of epistasis and the allelic direction across multiple metabolites (Please see the materials and methods for a more detailed description of this approach and *Figure 7—figure supplement 1* for a representative model plot).

To compare the role of each locus in a pairwise epistatic interaction, we plotted the phenotypic center for only the metabolites that showed significant epistasis for each pairwise interaction in our pairwise marker model analysis (*Figure 7* and *Figure 7—figure supplement 2*). For all cytonuclear epistatic interactions, the axis representing cytoplasmic genetic variation showed a wider phenotypic range than the axis representing variation in the nuclear locus (*Figure 7B,C* and *Figure 7—figure supplement 2*). Additionally, the range of phenotypic effects in epistasis between two nuclear loci was typically smaller than that between nuclear and cytoplasmic genome (*Figure 7—figure supplement 2*). Thus, variation within the cytoplasmic locus has a larger epistatic effect upon metabolomic variation than genetic variation in the nuclear locus further suggesting that genetic variation in the cytoplasm has larger than recognized phenotypic consequences.

## Quantifying the model improvement by including cytoplasmic variation

The above analysis showed that the cytoplasmic genetic variation plays an important role in mediating variation within the plant metabolome. To quantify how much the inclusion of the cytoplasm improved the genetic models in describing metabolomic variation, we analyzed our additive and pairwise epistasis model models with and without the cytoplasmic terms. For each metabolite for each model, we estimated the percent of variance explained by the significant terms in the model (*Figure 8*). This showed that including the cytoplasmic genome in the additive model nearly doubled the average variance compared to additive model containing only the nuclear loci. Adding the pairwise epistatic terms to the additive model without the cytoplasmic genome nearly tripled the fraction of metabolite variance explained by the genetic model without over-fitting the model (*Figure 8*). Including the cytoplasmic genome into the pairwise model shifted the average model genetic variance from 38% to 53% (*Figure 8*). Thus, the inclusion of the cytoplasmic term into the pairwise model explained as much (15%), if not more, variance than could be explained from the simple additive model built with only nuclear loci (13%) (*Figure 8*). This shows that the cytoplasmic genetic variation is critical to being able to fully describe metabolomic variation within Arabidopsis.

## Cytonuclear epistasis alters nuclear epistatic relationships

The epistatic network diagram showed that there were several instances of triangles where two nuclear loci showing pairwise epistasis also interacted with the cytoplasmic variation (*Figure 6*). This connectivity between the cytoplasmic genetic variation and nuclear QTL epistatic pairs led us to question if genetic variation in the cytoplasmic genomes could alter epistatic interactions between nuclear loci. To test if cytoplasmic genetic variation could alter nuclear epistasis, we identified all pairs of nuclear loci that interacted with each other and the cytoplasmic genome (*Figure 6*). These 10 three-way interactions were then included in the pairwise epistatic model to generate a three-way interaction model (*Figure 6—source data 1,2*). We found that including these 10 new three-way

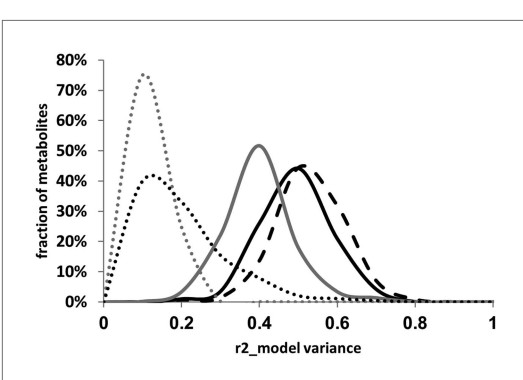

**Figure 8**. Distribution of estimated variance between main effects and epistatic interactions. The cytoplasmic term was added or dropped from the different statistical models to compare the total variance explained by each model for each metabolite. Dotted lines show estimated variance using solely the main effect loci without interactions (additive model), solid lines show the distribution of estimated variance across the metabolites using the pairwise epistasis model while the dashed lines are the results for the three-way epistasis model including the most prevalent three-way interactions as indicated by the epistatic network. The gray lines show the models with only the nuclear loci while the black lines show the model with the nuclear and cytoplasmic loci. For the frequency plot, bin size is 0.025 $r^2$.

interactions significantly increased the total variance explained by the model for the distribution of metabolites while only using less than 42% of the available degrees of freedom (*Figure 8*, *t*-test p<0.001).

To examine how cytoplasmic genetic variation influences nuclear epistasis we visualized all significant three-way interactions using the phenotypic center approach, (*Figure 9—figure supplements 1–3*). This showed that the cytoplasmic variation enhanced or even changed nuclear epistasis. Within the significant Cytoplasm × M.I.83 × M.IV.23 epistatic interaction, we identified larger effects of the nuclear loci than had been possible in the pairwise model (*Figures 7 and 9*, *Figure 9—figure supplements 1–3*). For example pyruvate and metabolite 227710 have quite large nuclear epistatic interactions that are only visible in one or the other cytoplasmic background (*Figure 9B,C*). Neither metabolite had a large nuclear epistasis due to comparisons averaging across the different cytoplasmic backgrounds (*Figure 7* vs *Figure 9*). In contrast, salicylic acid had different patterns of nuclear epistasis in the two cytoplasmic genetic backgrounds (*Figure 9D*). A similar pattern of the cytoplasmic variation altering nuclear epistasis was seen for other three-way interactions (*Figure 9—figure supplements 4–7*).

## Discussion

### Cytoplasmic genetic variation has a significant contribution to metabolic variation

Even though maternal contribution to complex phenotypes has long been suspected, most previous genomic surveys of natural variation using transcriptomics or metabolomics have either not had the capacity to directly assess the influence of cytoplasmic genetic variation or just ignore it completely. Thus, we used the reciprocal Kas × Tsu Arabidopsis RIL population to directly quantify the role of cytoplasmic genetic variation in quantitative variation of metabolomics traits. Our analysis showed that the cytoplasmic genome variation affected the phenotypic variation for over 80% of metabolites. The affected metabolites included key parts of central metabolism and some of the most specialized metabolites (*Figures 1, 5 and 6*). Interestingly, the combined variance of the nuclear genome was larger than the variance due to cytoplasmic genome (*Figure 1*). In contrast, cytoplasmic genetic variation affected more metabolites than any of the individual nuclear loci typically with larger effects (*Figures 3–5* and *Figure 4—figure supplements 1–5*). This is true even though the average nuclear locus spanned approximately the same number of genes as the organellar genomes combined.

We also identified a high level of significant epistatic interaction between cytoplasmic genetic variation and nuclear genetic variation, as cytonuclear epistasis (*Etterson et al., 2007*; *Tang et al., 2007*; *Wolf, 2009*). This cytonuclear epistasis explained as much, if not more, metabolic variation than the combined additive effects of the nuclear loci. In addition, the cytoplasmic genome controlled larger phenotypic changes than the nuclear locus within a cytonuclear interaction (*Figure 7*). This interaction between the cytoplasmic and nuclear loci was further extended to three-way interactions and showed that the cytoplasmic background could hide or alter the interaction between two nuclear loci (*Figure 9*). Thus, the cytoplasmic genetic background plays a key role in determining how natural variation within nuclear loci will function.

### What is the mechanistic basis of cytoplasmic genetic variation?

The effect of the cytoplasmic genetic variation was spread out across nearly all of primary metabolism making it impossible to ascribe a causal link with solely plastidic or mitochondrial genes (*Figure 5*). Similarly analysis of genomic variation in the two organelles found a large number of SNPs that could be affecting numerous genes within the organelles with an observed bias towards genes involved in the NADH dehydrogenase complex (*Figure 5—source data 1*). Thus, organellar genomes could be playing a role in the cytoplasmic genetic influence on metabolite variation, but given their coinheritance, it is not possible to resolve the direct causal polymorphisms. Separating the two genomes to resolve their relative roles would require identifying rare individuals where the two organelles show bi-parental inheritance instead of the uni-parental/maternal mode of inheritance typical for Arabidopsis (*Azhagiri and Maliga, 2007*).

The plant cytoplasmic genomes contain only about 1% of the number of genes as found within the nuclear genome (*Arabidopsis genome initiative, 2000*). Yet genetic variation in this small fraction of genes has a large consequence upon the plants metabolic variation (*Figure 5*). This could be from genetic polymorphisms in organellar genes that are central to plant metabolism, such as those central

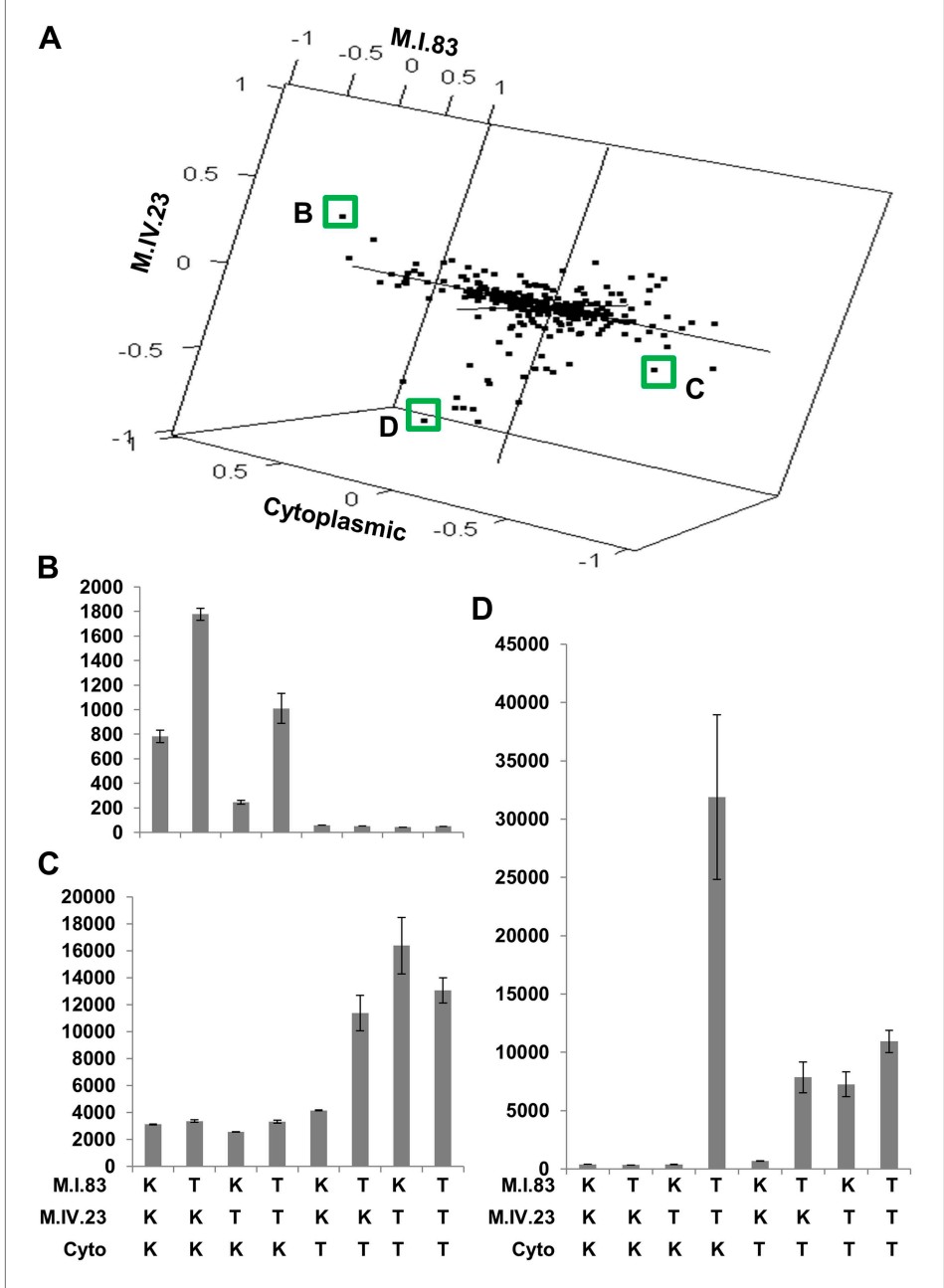

**Figure 9**. The cytoplasmic background alters nuclear epistatic interactions. The analysis of the M.IV.23 × M.I.83 × cytoplasmic three-way epistatic interaction is shown for all significantly affected metabolites. All other metabolite distributions for three-way epistatic combinations are plotted in *Figure 9—figure supplements 1–3*. (**A**) The center of mass calculations were used to estimate the phenotypic center for each metabolite that shows a significant three-way epistasis with the M.IV.23, M.I.83 and cytoplasmic genetic variation. All significant metabolites are plotted as unique points. The specific metabolites boxed and labeled show the location of the metabolites shown in parts, B, C and D respectively. For each locus, the Kas allele is plotted at 1 while Tsu is −1. (**B**) Effect of the M.IV.23 × M.I.83 × cytoplasmic epistasis upon the accumulation of unknown 227710. Average and standard error are shown. (**C**) Effect of the M.IV.23 × M.I.83 × cytoplasmic epistasis upon the accumulation of pyruvate. Average and standard error are shown. (**D**) Effect of the M.IV.23 × M.I.83 × cytoplasmic epistasis upon the accumulation of salicylic acid. Average and standard error are shown.

*Figure 9. Continued on next page*

*Figure 9. Continued*

The following figure supplements are available for figure 9:

**Figure supplement 1**. Distribution of phenotypic centers of metabolites significantly affected by M.I.83 × M.IV.3 × cytoplasmic three-way epistatic interaction.

**Figure supplement 2**. Distribution of phenotypic centers of metabolites significantly affected by M.I.83 × M.IV.17 × cytoplasmic three-way epistatic interaction.

**Figure supplement 3**. Distribution of phenotypic centers of metabolites significantly affected by M.I.83 × M.V.82 × cytoplasmic three-way epistatic interaction.

**Figure supplement 4**. Distribution of phenotypic centers of metabolites significantly affected by M.I.83 × M.IV.23 × cytoplasmic three-way epistatic interaction.

**Figure supplement 5**. Movement of phenotypic centers of metabolites significantly affected by M.I.83 × M.IV.23 × cytoplasmic three-way epistatic interaction.

**Figure supplement 6**. Long range movement of phenotypic centers of metabolites significantly affected by M.I.83 × M.IV.23 × cytoplasmic three-way epistatic interaction.

**Figure supplement 7**. Distribution of phenotypic center of metabolites significantly affected by M.I.83 × M.IV.23 × cytoplasmic three-way epistatic interaction.

to photosynthesis or NADH synthesis (*Figure 5—source data 1*). Alternatively, the polymorphisms could be within the organellar genes that are required to facilitate the function of the several thousand nuclear genes whose protein products are transported into the organelles and function there (*Ajjawi et al., 2010*). Finally, it is possible that the polymorphisms in the organelle impact retrograde signaling pathways, thus altering the function of the nuclear regulatory mechanisms (*Vinti et al., 2000*; *Larkin et al., 2003*; *Estavillo et al., 2011*; *Xiao et al., 2012*). Identifying the specific causal polymorphisms within the cytoplasmic genomes will require direct whole genome sequencing of the organellar genomes from all available genotypes to query the extent of natural variation in the organellar genome (*Cao et al., 2011*). An alternative may be to identify the causal genes underlying the nuclear loci involved in three-way interactions with the cytoplasm to triangulate the identity of the cytoplasmic gene.

## Conclusion

In this work, we report a first genomic survey of how genetic variation within cytoplasmic genomes influences metabolomic variation. Cytoplasmic genetic variation alters metabolite variation with effects that are equivalent to, if not greater than, individual nuclear loci. More importantly, the cytoplasmic background significantly influences the ability to detect epistasis between nuclear loci. Thus cytoplasmic genomes should be included in any future analysis of natural variation, either by being included as a genotype in GWA studies or by designing future populations as reciprocal populations to allow for direct analysis of the cytoplasmic genomic variation in controlling the phenotype. This inclusion will allow direct assessment of how cytoplasmic genomic variation influences other phenotypic classes, such as transcriptomics or broader physiological phenotypes. Natural genetic variation in the organellar genomes while frequently ignored will have to be kept at the front of future experimental approaches designed to understand the evolution and genetic architecture of organismal phenotypes.

## Materials and methods

### Growth of the Kas × Tsu RIL population

Seeds of the 341 lines of the Kas × Tsu recombinant inbred population were obtained from the Arabidopsis Biological Resource Center (ABRC, Columbus OH, USA) (*Juenger et al., 2006*, *2010*; *McKay et al., 2008*). We grew a total of four to five plants per line, split into two randomized complete blocks per experiment with two independent experiments separated by approximately

3 months. This provides four independent replicates per genotype. We grew plants in large planting trays with 156 individual wells (b × w × h: 30 × 25 × 100 mm), filled with standard potting soil (Sunshine Mix #1, Sun Gro Horticulture, Bellevue WA). Prior to sowing, we imbibed seeds in distilled water and cold stratified them at 4°C for 4 days. We placed approximately 3–5 seeds of a single genotype in the center of a well and covered the trays with a transparent plastic hood to retain humidity during germination. Plants from both reciprocal subpopulations where intermixed in the block design to allow direct statistical testing of cytoplasmic effects. After 1 week, we removed the transparent hoods and surplus plants to leave one seedling per well. We watered plants twice a week with nutrient-enriched water (0.5% N-P-K fertilizer in a 2-1-2 ratio, Grow More 4-18-38, Grow More Inc., Gardena CA) and kept them in a climate-controlled chamber at 22°C and a day/night cycle of 10 hr/14 hr. These plants and experiment were the same as previously described (*Joseph et al., 2013*).

## Metabolomics analysis

31 days after sowing, we harvested plants for metabolomics analysis. One leaf from the first fully mature adult leaf pair of each plant was removed and ground in extraction solution as previously described (*Rowe et al., 2008*; *Chan et al., 2010*). Metabolite identity was determined by comparing retention time and mass to the 2007 UC Davis Genome Center Metabolomics Facility metabolites database (http://fiehnlab.ucdavis.edu/Metabolite-Library-2007; *Fiehn et al., 2005*). Mixed samples were run approximately every 20 samples to optimize the peak identification and quantification algorithms and to control for variation in the detection as previously described (*Fiehn et al., 2008*; *Fernie et al., 2011*). The ion count values were used as a surrogate for metabolite abundance. Metabolite abundance was median normalized prior to analysis to account for any technical variation between samples. A separate leaf was extracted for glucosinolates and analyzed by HPLC according to previously described methods with the results reported elsewhere (*Kliebenstein et al., 2001a, b*). From the analysis of glucosinolates, a set of 10 lines with aberrant or genetically impossible glucosinolate profiles based on the known parentage were removed from the analysis, leaving a total 316 lines.

## Estimation of heritability

All RIL lines were represented in every block in both experiments creating a perfectly balanced randomized complete block design. All phenotypic data was used to calculate estimates of broad-sense heritability (H) for each phenotype as $H = \sigma^2_g/\sigma^2_p$, where $\sigma^2_g$ was estimated for both the RIL genotypes and cytoplasmic genotypes and $\sigma^2_p$ was the total phenotypic variance for a trait (*Liu, 1998*). The ANOVA model (Line heritability Model) for each metabolite phenotype in each line ($y_{gmeb}$) was: $y_{gceb} = \mu + C_c + G_g(C_c) + E_e + B_b(E_e) + C_c \times E_e + \varepsilon_{gceb}$, where c = the Kas or Tsu cytoplasm; g = the 1…316 for the 316 RILs, e = experiment 1 or 2 and b = block 1…8 nested within experiment. This allowed cytoplasmic effects to be directly tested in the C term and each RIL genotype (G) nested within the appropriate cytoplasmic class, either Kas or Tsu. Experiment and block nested within experiment were treated as random terms within the model to better parse the variation. All resulting variance estimates, p values and heritability terms are presented (*Figure 1—source data 1*). $\sigma^2_g$ for RIL was pulled from the $G_g(C_c)$ term while $\sigma^2_g$ for cytoplasmic variation was pulled directly from the $C_cM_m$ term. We used mean values for the RILs for further analysis as we had a randomized complete block design with no missing lines. Additionally, means and LSmeans were correlated with an average $r^2$ of 0.96 for the 159 known metabolites present in both experiments.

## QTL analysis

We the previously reported genetic map for these lines of the Kas × Tsu RIL population (*McKay et al., 2008*; *Joseph et al., 2013*). To detect metabolite QTLs, we used the average phenotype per RIL across all experiments (*Figure 1—source data 2*) (*Basten et al., 1999*; *Zeng et al., 1999*; *Wang et al., 2006*). For QTL detection, composite interval mapping (CIM) was implemented using cim function in R/qtl package with a 10 cM window. Forward regression was used to identify three cofactors per trait. The declaration of statistically significant QTLs was based on permutation-derived empirical thresholds using 1000 permutations for each mapped trait. QTLs with a LOD score above 2 were considered significant for further analysis (*Churchill and Doerge, 1994*; *Doerge and Churchill, 1996*). Composite interval mapping to assign significance based on the underlying trait distribution is robust at handling normal or near normal trait distributions (*Rebai, 1997*), as found for most of our phenotypes.

The define.peak function implemented in R/eqtl package was used to identify the peak location and one-LOD interval of each significant QTL for each trait (*Wang et al., 2006*). The effectscan function in R/qtl package was used to estimate the QTL additive effect (*R Development Core Team, 2012*). Allelic effects for each significant QTL are presented as percent effect, by estimating $[\overline{x}_{Tsu} - \overline{x}_{Kas}]/\overline{x}_{RIL}$ for each significant main effect marker.

QTL clusters were identified using a QTL summation approach where the position of each QTL for each trait was plotted on the chromosome by placing a 1 at the peak of the QTL. This was then used to sum the number of traits that had a detected QTL at a given position using a 5 cM sliding window across the genome (*Kliebenstein et al., 2006*). The QTL clusters identified defined genetic positions that were named respective to their phenotypic class and genetic positions with a prefix indicating the phenotype followed by the chromosome number and the cM position. For example, M.I.83 indicates a metabolomics QTL hotspot on chromosome I at 83 cM. The QTLs detected at the previously characterized and cloned glucosinolate AOP locus lies underneath the M.IV.17 metabolomics hotspot (*Magrath et al., 1994*; *Kliebenstein et al., 2001c, a*; *Kroymann et al., 2003*).

## Additive ANOVA model

To directly test the additive effect of each identified QTL cluster we used an ANOVA model containing the markers most closely associated with each of the significant QTL clusters as individual main effect terms. For each metabolite the average accumulation in lines of genotype $g$ at marker $m$ was shown as $y_{gm}$.

The model (Additive Model) for each metabolite in each line ($y_{gm}$) was: $y_{gm} = \mu + \sum_{g=1}^{2}\sum_{m=1}^{m} M_{gm} + \varepsilon_{gm}$, where

$g$ = Kas(1) or Tsu(2); $m$ = 1, …,14. The main effect of the markers was denoted as $M$ involving 15 markers ($m$). The cytoplasmic genome was included as an additional marker to test for cytoplasmic genome effects. We tested all metabolites with the appropriate model using lm function implemented in the R/car package, which returned all p values, Type III sums-of-squares for the complete model and each main effect. QTL main-effect estimates (in terms of allelic substitution values) were estimated for each marker (*Fox and Weisberg, 2011*; *R Development Core Team, 2012*). There is no significant single marker or pairwise segregation distortion in this population indicating that the model is balanced for all markers (*McKay et al., 2008*).

## QTL epistasis analysis

To test directly for epistatic interactions between the detected QTLs, we conducted an ANOVA using the pairwise epistasis model. We used this pairwise epistasis model per metabolite because we had previous evidence that RIL populations have a significant false negative QTL detection issue and wanted to be inclusive of all possible significant loci (*Chan et al., 2011*). Within the model, we tested all possible pairwise interactions between the markers. For each phenotype, the average value in the RILs of genotype $g$ at marker $m$ was shown as $y_{gm}$. The model (Pairwise epistasis model) for each metabolite in each line ($y_{gm}$) was: $y_{gm} = \mu + \sum_{g=1}^{2}\sum_{m=1}^{m} M_{gm} + \sum_{g=1}^{2}\sum_{m=1}^{m}\sum_{n=m+1}^{m} M_{gm}M_{gm} + \varepsilon_{gmn}$ where $g$ = Kas(1) or Tsu(2);

$m$ = 1, …,14 and n was the identity of the second marker for an interaction. The main effect of the markers was denoted as $M$ having a model involving 15 markers. The cytoplasmic genome was included as an additional single-locus marker to test for interactions between the cytoplasmic and nuclear genomes. p values, Type III sums-of-squares for the complete model and each individual term and QTL pairwise-effect estimates in terms of allelic substitution values were obtained as described for additive model ANOVA (*Fox and Weisberg, 2011*; *R Development Core Team, 2012*). Significance values were corrected for multiple testing within a model using FDR (<0.05). The main effect and epistatic interactions of the loci were visualized using cytoscape.v2.8.3 with interactions significant for less than 10% of the phenotypes were excluded from the network analysis (*Rowe et al., 2008*; *Smoot et al., 2011*). The 10% threshold was chosen as an additional multiple testing correction to provide a more conservative image of the network. There are no pairwise locus segregation distortions within this population showing that the genotypes in this analysis are balanced (*McKay et al., 2008*). The same style of model was run to test for specific three-way interactions by including specific three-way terms as indicated (three-way epistasis model).

## Plotting the epistatic center of phenotype

To plot the epistatic effect of QTLs upon a set of metabolites, we utilized the center of mass calculations. To do this we transitioned the physical distance metric in center of mass to a genetic distance

metric for mapping the center of phenotype. In this, we classify one locus of each interaction as being on the x-axis and the other locus being positioned on the y-axis. On each axis, the allelic value of each specific genetic locus is plotted in relation to the heterozygote. For instance, in a two QTL situation the x-axis would be the alleles of QTL1 with the Kas having an allelic value of 1 and the Tsu having an allelic value of −1 while the other axis would be the same for QTL2 there by positioning each of the four homozygous genotypic class in one of the four quadrants. For each metabolite, the center of phenotype was calculated using center of mass calculations as $x_{geno} = \frac{\sum_{i=1}^{N} p_i x_i}{\sum_{i=1}^{N} p_i}$ and $y_{geno} = \frac{\sum_{i=1}^{N} p_i y_i}{\sum_{i=1}^{N} p_i}$

where $p$ is the average un-scaled phenotypic value for each of the four homozygous genotypic class and $x$ is the x-coordinate of the corresponding genotypic class and $y$ is the y-coordinate of corresponding genotypic class. The center of phenotype ($x_{geno}, y_{geno}$) of all the metabolites significant for an interaction were plotted to visualize the distribution centers of phenotype for each pairwise interaction.

## Sequence diversity between Kas and Tsu Organelles

Kas-1 and Tsu-1 reads were obtained from NCBI (www.ncbi.nlm.nih.gov/, accessions SRX246466 and SRX246442 respectively) and aligned using Bowtie2 (*Langmead et al., 2009*; *Langmead and Salzberg, 2012*) to the Col-0 mitochondrial and chloroplast references obtained from TAIR (www.arabidopsis. org/) (*Rhee et al., 2003*). Aligned sequence reads were subsequently processed using SAMtools (*Li et al., 2009*), Picard (http://picard.sourceforge.net) and the Genome Analysis Toolkit (GATK) (*McKenna et al., 2010*). SNP discovery between Kas-1 and Tsu-1 was carried out using the UnifiedGenotyper package of GATK (*Figure 5—source data 1*).

# Additional information

### Competing interests

DJK: Reviewing editor, *eLife*. The other authors declare that no competing interests exist.

### Funding

| Funder | Grant reference number | Author |
|---|---|---|
| National Science Foundation | DBI0820580 | Bindu Joseph, Jason A Corwin, Baohua Li, Suzi Atwell, Daniel J Kliebenstein |

The funder had no role in study design, data collection and interpretation, or the decision to submit the work for publication.

### Author contributions

BJ, JAC, Conception and design, Acquisition of data, Analysis and interpretation of data, Drafting or revising the article; BL, Acquisition of data, Analysis and interpretation of data, Drafting or revising the article; SA, Acquisition of data, Analysis and interpretation of data; DJK, Conception and design, Analysis and interpretation of data, Drafting or revising the article

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
