## [Decision Letter]

Thank you for sending your work entitled “Cytosolic genetic variation and extensive cytonuclear interactions control natural variation in the metabolome” for consideration at *eLife*. Your article has been favorably peer reviewed by Deputy editor Detlef Weigel and another reviewer.

The two reviewers discussed their comments before we reached this decision, and they have assembled the following comments to help you prepare a revised submission.

The main point of this paper, that cytoplasmic factors are likely to play an important role in phenotypic variation and should not be ignored, is important, and well substantiated in this paper. To support their claim with many phenotypes, the authors measured hundreds of metabolites in two *A. thaliana* recombinant inbred line populations, both from the same two accessions, but with one set having inherited the cytoplasm from one accession, and the other from the other accession. The authors convincingly show that inherited differences in the cytoplasm – almost certainly mutations in the mitochondrial and chloroplast genomes – have measurable and non-negligible effects on metabolites, and that there is substantial epistatic interaction between organellar and nuclear genomes.

That said, several claims seem overstated. Starting with the title, “control” is clearly too strong. First, the authors should spell out what fraction of the genetic variation is attributable to cytoplasmic variants, what fraction is due to C×G interactions, and how does this compare to straight G effects? Second, how representative the findings of the paper are of other phenotypes remains unknown, and the authors should be more circumspect when extending their claims to phenotypes that might have no or only a very indirect connection to metabolites. Furthermore, both reviewers found the paper hard to follow – the presentation needs to be greatly simplified, so that the important points come across better.

Specific criticisms:

1) The authors’ explanation for the average heritability due to the organellar genome being much lower than the one coming from the nuclear genomes seems reasonable. However, both reviewers had a hard time trying to understand why there are apparently no metabolites that show cytoplasmic heritability greater than 0.15. Two different models were used to estimate nuclear and cytoplasmic heritability, but why? Along these lines, how do cytoplasmic and nuclear heritability compare for the metabolites significant for both? And on a related note, please explain why significant effects of the nuclear genome were estimated by partitioning the variance across 316 RIL genomes. These are not independent: surely you want the estimates produced by linkage mapping? Finally, that the complete cytoplasmic genome has a larger effect than individual nuclear loci (Figure 6) does not seem surprising, given that the mtDNA and cpDNA represent contain dozens of genes.

2) A substantial criticism is that the authors do not tell us anything about sequence differences in the organellar genomes of the RIL parents. The short read data are available, so it would not be too much of a burden to determine how many coding variants there are, and how the distribution of coding variants compares between biosynthetic genes in the organellar and nuclear genomes. We note that the Discussion states that differences have been found in the organellar genomes.

3) You discuss whether RILs have less phenotypic variation because they have less genetic variation. You write “A common concern of RIL populations is that they only sample two alleles per locus and thus might be limited in their genetic variation relative to species-wide diversity (Kim et al., 2007; Nordborg and Weigel, 2008).” This may or may not be common concern, but it is not the chief one, in our opinion, and it is also not the point made in the papers cited. As far as one can tell, Kim et al. (2007) does not discuss this issue at all, and Nordborg and Weigel (2008) make a completely different point, namely that RILs have greater power to detect rare alleles of large effect that “may not be important from an evolutionary perspective”. This is central to your paper because of the very real possibility that the observed QTL hotspots (which mostly differ from another RIL population) are due to rare alleles of large effect, which may well be deleterious. This does not invalidate the results, but changes their interpretation, in particular with respect to their evolutionary significance.

4) There is little consideration of existing knowledge of the contribution of organellar biosynthesis steps to metabolic pathways. An important question is whether metabolites that are known to be made in organelles are more likely to be affected by variation in the organellar genomes than those that are not.

Finally, the authors make the important point that the organellar genotypes would add greatly to GWA studies. We agree. Since the authors have metabolome data for 96 accessions (13), one wonders how much effort it would be to download existing read data, identify polymorphisms in the mitochondrial and chloroplast genomes, and include them in the GWA analysis. (A major unknown is what the distribution of shared polymorphisms in the organellar genomes is, since GWA studies rely on linkage having been broken up by recombination.) Such an analysis would greatly elevate the study and the authors are strongly encouraged to consider this very seriously.

Overall, the clarity of the manuscript should be improved considerably. The text is replete with jargon and technical terms that are not properly introduced. That the models used are not always clearly stated also contributes to the paper being hard to read. The main model(s) need to be in the main text, so it is clear where the results come from.

---

## [Author Response]

*Starting with the title, “control” is clearly too strong*.

We have changed the title to reflect influence rather than control. Additionally, we have worked to better caveat the work throughout to reflect the link to quantitative analysis of organelles in a genomic phenotype.

*First, the authors should spell out what fraction of the genetic variation is attributable to cytoplasmic variants, what fraction is due to C×G interactions, and how does this compare to straight G effects*?

We have worked to make this clearer by dissecting all variance components into the individual main effect (nuclear v cytoplasmic) and epistatic (cytonuclear v nuclear x nuclear) and included this in the description to help with the interpretation of our results. This is also specifically shown in Figure 6–figure supplement 1 where we parse genetic variation to each specific term of main effect or interaction for all metabolites.

*Second, how representative the findings of the paper are of other phenotypes remains unknown, and the authors should be more circumspect when extending their claims to phenotypes that might have no or only a very indirect connection to metabolites*.

We have worked to better lay out that while this manuscript focuses on metabolomic variation, further work is required to test how this extends to other phenotypes. As a respectful point, we find it hard to imagine a phenotype that has no, or even an indirect connection to metabolites given the fundamental basis that metabolism plays in providing the energy and structure for any other phenotype to properly occur.

*Furthermore, both reviewers found the paper hard to followthe presentation needs to be greatly simplified, so that the important points come across better*.

We have worked to improve the writing and hope that this has been achieved.

*Specific criticisms*:

*1) The authors’ explanation for the average heritability due to the organellar genome being much lower than the one coming from the nuclear genomes seems reasonable. However, both reviewers had a hard time trying to understand why there are apparently no metabolites that show cytoplasmic heritability greater than 0.15. Two different models were used to estimate nuclear and cytoplasmic heritability, but why? Along these lines, how do cytoplasmic and nuclear heritability compare for the metabolites significant for both? And on a related note, please explain why significant effects of the nuclear genome were estimated by partitioning the variance across 316 RIL genomes. These are not independent: surely you want the estimates produced by linkage mapping? Finally, that the complete cytoplasmic genome has a larger effect than individual nuclear loci (*Figure 6*) does not seem surprising, given that the mtDNA and cpDNA represent contain dozens of genes*.

We are slightly confused by the first question as the analysis in question used a single model to estimate nuclear and cytoplasmic heritability that is shown in [Supplementary-material SD1-data]. This model uses the standard approach to estimate heritability in a RIL population where you use the proportion of line variation of the total variation to estimate genetic reproducibility. In a non-reciprocal RIL population the line variation accounts for the variance across all the lines in the population. We did this classical approach to provide a common entry point to the community into this analysis as well as a common linkage to most previous RIL analysis. We have also introduced a description saying that the second model is asking a different question than the per line heritability question asked by the model in [Supplementary-material SD1-data].

We are unsure as to why no metabolites had a cytoplasmic heritability above 0.15 using this analysis but this is what the data showed and by doing the classical approach we can now show that this number is misleading based upon the ensuing linkage mapping. To also help with the analysis, we have provided locus and interaction specific estimates of heritability (per locus/interaction R2) that is now discussed in the section on epistatic interactions. This analysis is shown in Figure 6–figure supplement 1. This analysis shows that if we parse the nuclear genetic variation into individual nuclear hotspot variation, the cytoplasmic heritability is actually greater on average than any specific nuclear hotspot.

Finally, we have also mentioned in the Discussion that the organellar genomes have roughly the same number of genes as an individual nuclear locus because given the genetic resolution of recombination these loci still represent a region of >100 genes.

*2) A substantial criticism is that the authors do not tell us anything about sequence differences in the organellar genomes of the RIL parents. The short read data are available, so it would not be too much of a burden to determine how many coding variants there are, and how the distribution of coding variants compares between biosynthetic genes in the organellar and nuclear genomes. We note that the Discussion states that differences have been found in the organellar genomes*.

We have obtained suitable short reads and provided a list of polymorphisms between the Kas and Tsu mitochondria and organelle as Figure 7–source data 1. This has been added into the section on cytoplasmic effects along with a commentary on how the changes are focused on NADH related metabolic genes, which would explain the widespread consequences in the metabolic grid of the organellar genetic variation because NADH is central to nitrate assimilation, the GOGAT cycle, lipid metabolism, and any process involving a cytochrome P450.

*3) You discuss whether RILs have less phenotypic variation because they have less genetic variation. You write “A common concern of RIL populations is that they only sample two alleles per locus and thus might be limited in their genetic variation relative to species-wide diversity (Kim et al., 2007; Nordborg and Weigel, 2008).” This may or may not be common concern, but it is not the chief one, in our opinion, and it is also not the point made in the papers cited. As far as one can tell, Kim et al. (2007) does not discuss this issue at all, and Nordborg and Weigel (2008) make a completely different point, namely that RILs have greater power to detect rare alleles of large effect that “may not be important from an evolutionary perspective”. This is central to your paper because of the very real possibility that the observed QTL hotspots (which mostly differ from another RIL population) are due to rare alleles of large effect, which may well be deleterious. This does not invalidate the results, but changes their interpretation, in particular with respect to their evolutionary significance*.

We have worked to better clarify this section to reflect both the power and rarity concern. The specific paragraph in question now attempts to show that while the RIL population may be susceptible to the rare allele issues, we have similar if not greater phenotypic variation than the accessions, meaning that we actually have better power to detect loci and interactions.

*4) There is little consideration of existing knowledge of the contribution of organellar biosynthesis steps to metabolic pathways. An important question is whether metabolites that are known to be made in organelles are more likely to be affected by variation in the organellar genomes than those that are not*.

We have added in a commentary in the first results section on the organellar variation of how most metabolites, even those specifically synthesized in a single compartment, are then transported throughout the plant cell even from one organelle to the other. Additionally, a number of metabolites are made in multiple compartments including the organelles and the cytosol, as such ascribing individual compartmental function is not as straightforward as is required for a hypergeometric test or the like. Additionally, our extraction protocol cannot differentiate between the different pools within a cell. Given that nearly all primary metabolites are influenced by the organellar genomic variation this analysis doesn’t have the power to refine the mechanistic basis of this change.

*Finally, the authors make the important point that the organellar genotypes would add greatly to GWA studies. We agree. Since the authors have metabolome data for 96 accessions (*[13]*), one wonders how much effort it would be to download existing read data, identify polymorphisms in the mitochondrial and chloroplast genomes, and include them in the GWA analysis. (A major unknown is what the distribution of shared polymorphisms in the organellar genomes is, since GWA studies rely on linkage having been broken up by recombination.) Such an analysis would greatly elevate the study and the authors are strongly encouraged to consider this very seriously*.

We agree that this would be nice, but for several reasons we feel that this is more appropriate for a follow-up study. The first is technical in that the available short read data were largely made with protocols to enrich the nuclear genomic DNA at the cost of the organellar sequences and as such, we had to identify specific Kas and Tsu samples prepped with a different protocol. The other available short read archive sequences did not provide us sufficient coverage on the organellar genomes to call polymorphisms. The second reason is genetic in that all published literature on the Arabidopsis organellar genetic variation using targeted gene analysis shows that the organelles do not show any evidence of independent assortment and instead they behave in a more clonal fashion which greatly complicates any potential for GWA as the entire organelles are essentially within complete LD. The final reason is for clarity, as the reviewers have rightly noted, we need to improve the manuscript’s clarity and adding a completely additional section on accessions and the organelles would require the equivalent of a full manuscript’s worth of analysis to assess the link between the organelle and population structure, LD, and then finally GWA with metabolites. Thus, we feel that this request is an excellent foundation for another complete manuscript, but it would not help to clarify this manuscript mechanistically, genetically, or editorially.

*Overall, the clarity of the manuscript should be improved considerably. The text is replete with jargon and technical terms that are not properly introduced. That the models used are not always clearly stated also contributes to the paper being hard to read. The main model(s) need to be in the main text, so it is clear where the results come from*.

We have worked throughout to better clarify the text and more explicitly link to the proper model.

[Editors’ note: after re-review, the authors edited the article further to clarify the primary message and removed sections of the article relating to comparative analysis to other populations.]